# Insecure Attachment, Oxytocinergic System and C-Tactile Fibers: An Integrative and Translational Pathophysiological Model of Fibromyalgia and Central Sensitivity Syndromes

**DOI:** 10.3390/biomedicines12081744

**Published:** 2024-08-02

**Authors:** Gianluca Bruti, Paola Foggetti

**Affiliations:** Eurekacademy, Center for International Studies of Cognitive Neurosciences and Integrated Medicine, Antonio Bertoloni 26/C, 00197 Rome, Italy

**Keywords:** fibromyalgia pathophysiology, stress response system, functional somatic disorders, central sensitized syndromes, central sensitization, insecure attachment style, oxytocinergic system, small-fiber neuropathy, C-tactile fibers, C-low-threshold mechanoreceptors, affective touch

## Abstract

Although the pathophysiology of fibromyalgia syndrome has been better understood in recent decades, a unified model of its pathogenesis and an effective therapeutic approach are still far from being realized. The main aim of this article will be to delve into the fundamental mechanisms of the pathophysiology of fibromyalgia conceptualized as stress intolerance syndrome. Using the biopsychosocial model of chronic pain syndromes, we will describe the potential role of the attachment system, C-tactile fibers, and oxytocinergic system dysfunction in the pathophysiology of fibromyalgia syndrome and other central sensitivity syndromes. At the end of the article, the therapeutic implications of this new global and translational pathophysiological model will be briefly discussed.

## 1. Introduction

In clinical practice, and also as clearly described by the American College of Rheumatology diagnostic criteria [1], fibromyalgia syndrome (FMS) represents a chronic and heterogeneous primary multi-symptomatic disorder. Table 1 shows all the symptoms which, with different phenotypic expression, can be clinically reported by patients suffering from FMS.

Indeed, although widespread pain, stiffness, cognitive impairment, interrupted and non-restorative sleep, and fatigue have been considered the core symptoms of FMS, a constellation of other significant complaints are often reported by FMS patients. In particular, the presence of anxiety and mood disorders are so important that for many years FMS was not considered a chronic pain syndrome per se, but somehow a sort of psychogenic pain which, at least in clinical psychiatric settings, it is still labeled as somatoform pain disorder or somatization disorder, according to the International Classification of Diseases [2].

More recently, the term “functional somatic disorders” has been introduced as a consequence of recent discoveries in the field of brain–body interaction and to resolve historical controversies between the somatic and mental nature of central sensitivity syndromes (CSSs) such as FMS [3].

The complexity of FMS is even more evident if we consider the multiple pathophysiological mechanisms that have been hypothesized to be responsible for the wide spectrum of symptoms reported by fibromyalgia patients, from anomalies of the hypothalamic–pituitary–adrenal axis and the autonomic nervous system, to dysregulation of the immune system; from the role of the central nervous system to that of the peripheral nervous system; and from genetic factors to environmental components and epigenetic mechanisms.

Probably, each of these dysfunctions can have a role in the pathogenesis of FMS, but we believe it is important from both a clinical and therapeutic point of view to identify the “primum movens” capable of connecting all these potential pathophysiological mechanisms together and considering FMS as belonging to the spectrum of CSSs, all disorders that share central sensitization as an underlying pathophysiological mechanism [4,5].

The aim of this narrative review is to report and analyze the main theories on the pathophysiology of FMS emerging from different fields of research focusing on the role of the bodily stress system (BSS) and to design a hypothetical and innovative pathophysiological model of FMS and of CSSs by integrating all the mechanisms that until now have been considered in the pathophysiology of CSSs.

We will begin by discussing and focusing on, in the first phase, the role of the body stress system in the pathophysiology of FMS and CSSs, demonstrating that the stress response system may already be altered before the presentation of the clinical syndrome.

After describing the physiological principles of the attachment system, the C-tactile fiber system and the oxytocinergic system, we will provide data to support the strategic role of these three systems and their mutual and dysfunctional associations in the pathophysiology of FMS and CSSs.

Finally, we will briefly discuss some therapeutic implications of this narrative review, summarizing in the conclusions what is also reported in the graphical abstract (Figure 1).

We hope that our new pathophysiological model, potentially common to different types of chronic pain syndromes, can promote scientific research in this field and a more compassionate way of conducting the doctor–patient relationship, thus leading to a more effective clinical approach.

## 2. Pathophysiology of Fibromyalgia Syndrome (FMS)

### 2.1. The Stress Response System in FMS and Central Sensitivity Syndromes (CSSs)

Although clinical observations and scientific findings support the theory of FMS as a stress-related disorder due to dysregulation of the bodily stress system (BSS) [6,7,8,9], the true origin of these pathophysiological mechanisms is still the subject of different interpretations [10,11].

From this perspective, many studies conducted on FMS have demonstrated that the two main players in the human stress system, the hypothalamic–pituitary–adrenal (HPA) axis and the sympathetic nervous system, present different types of dysfunctions, with some authors arguing for hypoactivity and others for hyperactivity of both systems [12].

The nature of these contradictory results probably depends on the complexity of human-stress neural circuits, which, to adapt the body to stress stimuli, work together in a mutual and synchronized way and influence each other with different positive feedback loops [13]. Moreover, the heterogeneity of the study design with regards to the selection of the clinical sample (age of the patients and duration of the disease), the methods applied for the evaluation of the BSS, and the different phases of the disease considered in the study, were considered the main factors responsible for these inconsistent findings observed in the pathophysiology of FMS [10,14].

Regardless, to explain this different pattern of BSS activation, it has been hypothesized that, over time, an increase in basal tone of both the HPA axis and sympathetic nervous system activity in the early stages of the disease (hyperactivity phase), could progressively predispose the patient to a breakdown in the acute stress-related response (hypoactivity phase), during which the ability to cope with daily stressful stimuli could be compromised [6,7,10,14,15].

From this perspective, and in line with previous results [14], more recently it has been demonstrated that in response to intense pain, the concentration of cortisol in the hair is higher in the early stages of FMS, while it tends to decrease afterwards, suggesting some sort of downregulation of cortisol during disease progression [16].

Certainly, these findings do not appear to be specific to FMS, but common to the spectrum of chronic overlapping pain conditions and functional somatic disorders (FSDs) [4]. Indeed, it has been theorized that the comorbidity between these overlapping clinical syndromes could depend on the same anomalies in the stress regulation system [6,7,17].

One of the most important methods used to evaluate the autonomic nervous system (ANS) in patients with fibromyalgia (FM) is cardiovascular regulation.

A large body of evidence suggests that in patients with FSDs, and in particular in patients with FM, basal heart rate variability (HRV) is lower than in healthy subjects, supporting the potential use of this cardiac parameter as a biological marker of chronic pain (CP) [18,19]. In fact, HRV and its sympathetic and parasympathetic components represent a reliable clinical tool for evaluating the body’s ability to cope with acute stress stimuli such as nociceptive stimulation [20], and for this reason it is also considered a biological indicator of human resilience [21].

Recently, the association between reduced HRV and FSDs was further found in a large-scale study suggesting a predominant role of the sympathetic arm of the ANS in the pathophysiology of CP [22].

The “chronic pain as ANS dysfunction” paradigm has been particularly demonstrated in FMS, characterized by a general deterioration of the stress response evidenced by increased basal sympathetic activity, a reduction in sympathetic reactivity to acute stress, and a reduction in parasympathetic tone [9,23,24,25,26,27].

In a recent clinical study, lower HRV and a more-impaired response to cognitive stress were found to be associated with mood and anxiety disorders in patients with FM, supporting a sharing of pathophysiological mechanisms between CP, psychiatric disorders and dysfunction of the autonomic nervous system [28].

Interestingly, the role of human stress dysfunction in the pathophysiology of FSDs was also confirmed in a general-population study which led to a unifying diagnostic construct termed bodily distress syndrome, which may be common to several clinical syndromes [29,30]. The importance of this definition lies in its prognostic implications as highlighted by a prospective study which shows how, in bodily distress syndrome, a deficit in response to psychological stress can predispose the patient to the development of diseases over time [31].

From this perspective, in the Hyland (2017) model [32], FMS and CSSs were considered the consequence of a maladaptive response to stimuli internal and external to the body, secondary to the organism’s inability to transform a “stop signal” into an adaptive behavior [32,33].

In other words, in this theory, clinical symptomatology, rather than a direct consequence of a pathogen, would represent a “stop signal” capable of orienting the behavior towards a target in order to cope with the potential threat to survival [33]. Over time, this inability would progressively increase the intensity of the “stop signal” and the related clinical symptom, determining an increasingly higher tone and state of symptomatology called a “stop programme” from which the clinical spectrum of CSSs would derive.

According to the Hyland model (2017) [32] the human body is metaphorically compared to a computer composed of hardware and software which can be damaged during use but which can also be repaired using a multimodal therapeutic approach called “body reprogramming” [32,33]. This non-pharmacological multimodal therapeutic approach, based on the three fundamental principles (“reduce the impact of stop signals”; “teach the body to experience “safety”; and “support the body’s hardware and software”) has been recently proven effective in a preliminary study conducted on a sample of patients (n = 198) with FMS and CSSs [34]. In this demonstration study, after eight treatment sessions each lasting 2.5 h, one per week, the group-based brain reprogramming course was effective in reducing symptom interference, anxiety and depression and in improving the overall quality of life in this clinical sample. Notably, the clinical benefit was observed not only at the end of the study, but also at a 3-months follow-up [34], suggesting that this multimodal approach is able to reshape maladaptive networks by converting them back into adaptive networks.

On the other hand, as clearly reported by the International Association for the Study of Pain [35], pain cannot be considered a pure sensation, but rather a stressful and unpleasant experience, characterized by sensory, emotional and cognitive components linked to the perception of real damage or a threat of it. This definition highlights the cognitive aspect of pain perception and modulation and the significance of pain as a threat to survival.

With this in mind, a recent pathophysiological and integrative model called “Fibromyalgia: Imbalance of Threat and Soothing Systems” has been proposed [36]. According to this model, patients with FMS could have an excessive functionality of the threat system and the related defense cascade which, over time, could predispose the patient to, and/or maintain, the clinical phenotype of FMS [36].

### 2.2. The Bodily Distress Syndrome and Fibromyalgia Syndrome (FMS): What Came First?

Although dysregulation of the bodily stress system (BSS) could explain the shared symptoms in FMS and other central sensitivity syndromes (CSSs), it is still a matter of debate whether these neuroendocrine abnormalities represent the cause or consequence of chronic pain (CP) and FMS [11,13,22,36,37].

A large body of scientific evidence supports the hypothesis of an association between trauma, both physical and psychological, and the onset of the first clinical manifestation of FMS, in particular when the trauma is emotional [38]. From this point of view, at least to a certain extent and in the initial phase of FMS, it is intuitive to consider the role of the BSS as primary and not secondary to the onset of pain.

Arguably, scientific discoveries emerging from studying the relationship between early life experiences and developmental psychobiology could be the key to answering what comes first with respect to BSS abnormalities and CP [39,40]. In fact, numerous findings from both animal and human research have widely demonstrated that early stressful life events may be responsible for maladaptive neural plasticity and function and that these abnormalities may have a negative impact on children’s psychological and physical health throughout life [39].

From this perspective, it has been shown that subjects with a positive history for childhood trauma are twice as likely to develop a CP adulthood (odds for fibromyalgia = 2.52; 95% CI = 1.92–3.31) [41]. Interestingly, the likelihood of developing functional somatic disorders is tripled when both childhood and adult trauma are included in the meta-analyses, supporting the key role of BSS in the pathophysiology of CSSs [42].

As previously mentioned, comorbidity between FMS and depressive and anxiety disorders is more the rule than the exception, and a positive history of childhood maltreatment has been considered to be a pathophysiological missing link of comorbidity between CP and psychiatric disorders. In particular, child maltreatment has been found to be highly associated with post-traumatic stress symptoms, a clinical spectrum that represents a frequent consequence of childhood maltreatment and which can also be considered a symptomatic phenotype predisposing the patient to CP in childhood and adulthood [43,44]. Interestingly, adverse experiences in childhood have also been found to be associated with CP in youth, suggesting a clear cause-and-effect relationship between early stressful life events and the pathogenesis of CP [45]. From this point of view, it has been shown that early stress and in particular childhood maltreatment, is highly associated with the hypothalamic–pituitary–adrenal (HPA) axis and autonomic nervous-system dysfunction, as well as alterations of other peripheral stress-response systems such as the immune and inflammatory ones [39,46]. According to this pathophysiological “early trauma model”, and considering the frequent clinical overlap between post-traumatic stress disorder (PTSD) and FMS (the prevalence varies between 56%), it has been suggested that all patients with fibromyalgia (FM) should be screened for the presence of PTSD [47]. The importance of these epidemiological and clinical findings is linked to the negative impact on therapy and prognosis that the presence of PTSD and also partial or subthreshold forms (post-traumatic stress symptoms) can have in patients suffering from FM [47,48]. In this regard, it has been shown that the level of central sensitization, measured as widespread pain, pain intensity and multisensory complaints, could depend even more on the symptoms of PTSD than on the intensity of exposure to trauma itself, suggesting an important role of this psychiatric syndrome in the pathogenesis of CP [48]. Furthermore, as with other CSSs, FMS and PTSD may share some pathophysiological mechanisms such as HPA axis dysregulation (significantly low basal cortisol) [49].

### 2.3. General Principles of the Attachment System (AS) Theory and Relationship with the Bodily Stress System (BSS)

According to John Bowlby (1969) [50], the father of attachment theory, attachment can be defined as a biobehavioral state composed of biological, physiological and behavioral systems, capable of maintaining internal homeostasis in the event of emotional arousal. In particular, the attachment system (AS) is a neurophysiological function characterized by various social behaviors such us taking care of children and eliciting care from others. In fact, attachment is an innate motivational system capable of forming a lasting, selective and affectionate bond with the primary caregiver ready to recognize, evaluate and satisfy the child’s non-verbal needs and requests. Consequently, parental sensitivity is crucial in determining the quality of the child’s AS and subsequent interpersonal relationships in adult life. Indeed, if the type of AS of the child depends on the behavior of the parents, it is also true that the latter is shaped by the behavior of the child. This bidirectional relational conditioning constitutes the framework on which the child’s self-protective behavioral strategies are based, together with the formation of attachment representations. Specifically, in human infants, the AS is activated during stressful situations. In other words, when the “threat system” is activated, the child’s behavior acts to evoke nurturing behaviors in the attachment figure in order to reduce the stress load and activate the child’s “soothing system” [50,51,52,53]. The mechanism that regulates this interpersonal biological process is called “biobehavioral synchrony” and is the basis of all human social and affiliative systems [54]. Over time, thanks to this bidirectional conditioning within the parent–child dyad, the child learns to self-soothe in cases of discomfort. These internal psychological, cognitive and behavioral representations of relaxation and comfort will constitute the “internal working models” that will be used consciously or unconsciously throughout adult life. Naturally, depending on the degree of “harmony” of the parent–child dyad, the child will have a secure or insecure attachment. In particular, adult behavior will be influenced by childhood representations of beliefs about being worthy of care (self-model) and whether others are worthy of his trust (other-model) (Figure 2) [55,56].

According to this model, several researchers consider the AS in itself to be a device responsible for regulating emotions [57]. From this perspective, the identikit of a person with SA is characterized by competence in social behavior, emotion regulation, and executive cognitive functions, with a balance between exploration and attachment behavior and an ability to analyze one’s feelings and thoughts. It has been shown that having a secure attachment (SA) predisposes one to having high-quality relationships and at the same time to positive health outcomes. On the contrary, while the ambivalent/preoccupied form of insecure attachment (IA) is characterized by feelings of anger and less autonomy, the typical aspect of the avoidant/dismissing form of IA is to deactivate the AS to avoid discomfort; in both forms of IA, poor emotional regulation leads to an increase in the response of the stress system to stimuli related to the AS (Figure 2) [58].

Over the last three decades, numerous studies have been published on the neurobiological and physiological mechanisms of the AS, suggesting that SA can be considered an indicator of resilience [54]. In fact, it has been shown that the activation of the AS in adult subjects with SA increases heart rate variability (HRV) (one of the most important biomarkers of resilience) through the parasympathetic response to stress, explaining the psychophysiological benefits deriving from having positive social support [59]. These findings were also recently replicated in adolescents with SA who demonstrated higher HRV compared to the IA groups when subjected to the Adult Attachment Projective System interview, suggesting once again that secure adolescents are more efficient in dealing with stressors arising from attachment representations [58]. In fact, it has been demonstrated that attachment relationships have a regulatory role on the physiological and psychological response to stress through the modulation of the bodily stress system (BSS) [60]. Consequently, compared to other attachment groups, the secure ones are able to respond with a lower increase in cortisol, lower skin conductivity and more flexible prefrontal functions during attachment-related stimuli [60].

On the other hand, numerous evidence supports the harmful role of IA on the biological, psychological and sociological aspects of adulthood. How can we explain these scientific findings?

It has been shown that people with IA, both infants and adults, are characterized by an augmented activity of the stress response system, supporting the paradigm of IA as a deficit of emotion regulation. In particular, a large body of evidence has demonstrated that early childhood adversity and/or poor parenting are associated with hypothalamic–pituitary–adrenal (HPA) axis hyper- or hypo-responsiveness, sympathetic–adrenal-medullary dysfunction, and immune-system dysregulations in infants, children, and adults with IA [61]. The association between dysfunctional neuroendocrine and immunological responses observed in IA subjects is not surprising, considering the reciprocal modulation that HPA, sympathetic–adrenal-medullary axes and the immune system exert on each other [62]. On the other hand, the lower HRV observed in adolescents with insecure avoidant attachment and unresolved attachment compared to that of adolescents with SA confirmed their reduced ability to regulate the autonomic nervous system in response to stressors related to the representations of attachment [58].

For the data reported above, BSS dysregulation could represent the missing link between having an IA during childhood and negative health outcomes such as psychiatric, cardiovascular, metabolic and tumor diseases in adulthood [61,63,64].

Thus, considering the importance of the AS on the regulation of BSS and affective states, IA is considered a psychobiological indicator of vulnerability to stress and may represent an important risk factor for several diseases, including pain [65,66,67].

### 2.4. The Relationship between Insecure Attachment (IA) in Chronic Pain (CP) and Fibromyalgia Syndrome (FMS)

Several studies have shown that insecure attachment (IA) represents not only a predisposing factor for chronic pain (CP) but also an important cause of altered pain perception, such as catastrophic pain, lack of ability to cope with pain and a significant predictive factor for anxiety and depression in CP disorders [65]. In reality, inconsistent results have been found on the relationship between IA and pain intensity, although numerous studies suggest that people with IA, in addition to being more prone to developing pain, perceive it as more pervasive, distressing and disabling [67,68].

In a cross-sectional study conducted on a large general-population sample, subjects with chronic widespread pain presented IA (preoccupied, dismissing, fearful) more frequently than subjects without pain. Interestingly, the number of pain sites was associated with an IA, along with more disabling pain symptoms [68]. Despite the cross-sectional nature of the study, and also considering the results found by other researchers, the authors concluded that people with an IA may be twice as likely to have chronic widespread pain. In fact, it has been found that the origin of the psychological trait “pain related to fear” is to be considered a consequence of IA, and in particular of its model of self-dimension (degree of anxiety regarding rejection based on beliefs of personal unworthiness) [69]. From this perspective, the old theory of CP as derived from a stress diathesis, has been transformed into attachment-diathesis models of CP, suggesting a direct and causal pathophysiological link between IA, stress and CP disorders [65]. An important confirmation of IA as the main source of a more-negative subjective perception of the painful experience came from studies conducted with experimentally induced pain [70].

As mentioned above, it has been shown that the negative effect of IA on the functioning of patients with CP may not be mediated by an increase in pain intensity, suggesting that other variables, like cognitive (catastrophizing, coping style) and affective (anxiety and depression) components of the pain experience, and social status such as marital satisfaction and social support from significant others, may have a role [67,68].

In the context of FMS, it has been proposed that attachment style should be regularly explored in patients with fibromyalgia, confirming the validity of “the attachment-diathesis model of chronic pain” [71]. In fact, as also found in other CP syndromes, IA is frequently observed in patients with FMS and appears not to be specific of this CP disorder [71,72]. Rather, it has been hypothesized that IA represents an important predisposing factor to alexithymia which, conversely, has been found to be a predictive personality trait of FMS [72,73]. Regardless, IA represents an important predictor of lower quality of life in patients with FMSm and this effect appears to be mediated by low self-esteem [74] and depressive symptoms [75]. In other words, the presence of IA (dismissive, preoccupied and fearful), could be responsible for a greater biopsychosocial impact and a higher disease burden in female patients with FMS [74,75].

### 2.5. History and Physiology of C-Tactile (CT) Afferents and Their Role in “The Affective Touch Hypothesis”

Over the past three decades, a critical step in the study of the somatosensory system has been the discovery of a class of unmyelinated cutaneous mechanosensory fibers called C-low-threshold mechanoreceptor (C-LTMRs) afferents [76,77]. The first evidence of the presence of this type of C fiber also in humans, renamed C-tactile (CT) fibers, emerged in a microneurography study conducted on the infraorbital trigeminal nerve, which was found to be highly responsive to gentle touch [78]. A few years later, this finding was further supported by a similar study performed on the supraorbital trigeminal nerve, which clearly demonstrated the different stimulation pattern between CT and nociceptive afferents [79]. Due to their small diameter and unmyelinated axon, CT fibers have a very slow conduction speed (between 0.3 and 2 m/s) and this delayed response is a useful neurophysiological parameter for easily recognizing the CT fiber signal. Furthermore, CT fibers, by definition, have a low mechanical-stimulation threshold with a monofilament stimulus response ranging from 5 mN to 0.04 mN (~500 mg–~4 mg) (Table 2) [80]. However, it has been shown that some C nociceptors are even able to respond after a monofilament stimulus with a low mechanical-stimulation threshold (~2.5 mN). But while CT fibers respond to low-intensity mechanical stimulation with a burst of spikes, the response of C nociceptors can produce, at most, a couple of spikes. CT afferents are abundant in hairy skin, and they can be activated when a tactile stimulus, such as a skin-stroking caress, is applied to the skin at a specific speed (3–6 cm/s), with a determined pressure (0.04–5 millinewton) and with an optimal temperature (skin temperature) [80,81]. The general neurophysiological characteristics of CT fibers are summarized in Table 2 [80].

Indeed, it was only two decades after their first discovery that “the affective touch hypothesis” was formulated [81]. In this model, “the essential role of the C-tactile system is to convey pleasant aspects of light touch, especially skin-to-skin contact with affiliated humans”.

It is interesting to note that, starting from this hypothesis, several research groups have stated that, as in the pain system, in the sensory system also there are central tactile discriminative and affective representations that depend on the activity of the Aβ and CT afferents, respectively [82].

### 2.6. C-Tactile (CT) Fibers in Fibromyalgia Syndrome (FMS): From Small-Fiber Pathology (SFP) to Pain Modulation through the Stimulation of “The Affective System C”

Although there is evidence of an important association between fibromyalgia syndrome (FMS) and small-fiber pathology (SFP), the pathophysiological mechanism underlying this comorbidity is still unclear. Indeed, SFP has been shown to be present in a proportion of nearly 50% of patients with FMS, supporting a role for peripheral mechanisms in the pathophysiology of fibromyalgia (FM), at least in a broad range of patients [83]. On the other hand, with the increasing number of studies on intraepidermal nerve fiber density (IENFD) (65% in human studies and 440% in non-human studies from 2010 to 2020), a reduction in IENFD was, surprisingly, found in various human diseases [84]. In any case, on the one hand it is not yet clear whether these findings could be the cause or consequence of FMS, and on the other hand, there is an urgent need to discover the common mechanisms underlying the loss of IENFD observed in different types of diseases [83,84,85]. Interestingly, in an experimental model of pain in rats, it has been demonstrated that the bilateral increase in glutamatergic activity at the level of the insular cortex is responsible for both the multimodal pain behavior and the loss of peripheral fibers. According to their results, the authors suggested that an overactivity of the insular cortex could be the main cause of the pathogenesis of SFP in patients with FM [86].

In contrast to the theory of centralization of pain in FM, some authors have hypothesized that the dorsal roots of the spinal ganglia (DRG) could play a role in the genesis of hyperalgesia in FMS starting from various stressful stimuli, including the psychological ones. From this point of view, the alteration of small peripheral fibers has been considered the most important evidence of the link between FMS and DRG pathology [87]. In favor of the role of DRG pathology and peripheral sensitization as pathophysiological mechanisms of FMS, it has been demonstrated that IgG from FMS patients, beyond the ability to bind to human DRG, when inoculated into mice, is able to produce painful sensory hypersensitivity by sensitizing peripheral nociceptive afferents and causing a reduction in IENFD [88]. According to other authors, a high sympathetic activity could be responsible for an increased response of the immune system which, together with the high level of muscular tension and the related myofascial-derived antigen anomalies, could lead to the formation of immune complexes capable of creating hyperexcitability in the DRG [89].

Moreover, it has been argued that, in the absence of severe depression and anxiety, the clinical phenotype of FMS and its associated disease burden correspond to that of neuropathic pain with dysautonomia. In this model, mood and anxiety disorders would not only not be clinical indicators of pain centralization, but should also be considered confounders of the peripheral source of sensitization [90]. Indeed, the difficulty in interpreting the results stated above is even more pronounced considering a recent study, which demonstrated the presence of a subgroup of patients with FMS who, without any alteration of nociceptive afferents, showed the clinical phenotype of small-fiber neuropathy [85]. On the other hand, in diabetic neuropathy, a positive correlation between the loss of IENFD and the presence of pain has not been found, and for this reason the loss of fibers is not a reliable predictor of the development of neuropathic pain, even in patients with diabetes [84].

Although the presence of SFP in FMS has recently been confirmed in a large percentage of patients, it has been clarified that SFP observed in FMS must be differentiated from small-fiber neuropathy, being different from it not only from the neurophysiological point of view, but also from the point of view of the clinical phenotype [83,91,92].

How can we interpret such a large amount of SFP in FMS, and what kind of role could it have from a pathophysiological perspective?

Interestingly, the involvement of the opioid system, both in the pathophysiology of FMS and in the physiology of the pleasant sensation due to the activation of C-tactile (CT) afferents, has been found in patients suffering from FM, confirming an alteration of the opioid system in this chronic pain (CP) condition and linking it, for the first time, to an abnormality of the function of CT afferents [93]. A confirmation of the involvement of a dysfunction of the CT afferents and the related affective tactile system was corroborated in a more recent neuroimaging study in FMS. In this study, the authors suggest that the reduction in the activity of the posterior part of the insular cortex during the pleasant rating, and the activation of its anterior region during pain assessment, could be linked to normal sensory processing associated with anhedonia to pleasant touch and abnormal evaluative processing, confirming the potential role of CT afferents in the physiology and pathophysiology of both acute pain and CP modulation [94].

However, it still remains to be clarified which peripheral and central mechanisms underlie the dysfunction of CT afferents and, consequently, the related lack of pleasant perception.

Furthermore, the enigmatic physiology of this C sensory system is based on its potential pro-nociceptive and non-nociceptive properties [95]. It has, in fact, been demonstrated that C-LTMRs afferent stimulation is able to potentially produce not only a reduction in pain but also a type of allodynia which, compared to that mediated by Aß, is less topographically correlated to the side of the damage. From this point of view, C-LTMRs-mediated allodynia resembles that typically observed in patients with widespread pain such as FMS [95].

On the other hand, since the activation of CT afferents is associated with a subjective sensation of pleasantness, a series of studies have been conducted on acute pain induced in healthy controls, demonstrating the analgesic power of gentle touch [96,97].

The mechanism of action underlying the analgesic effect of stimulation of CT afferents has been studied in animal models and in human clinical studies. In particular, in animal models, CT afferents would act, on the one hand, on the substantia gelatinosa localized in the dorsal horn of the spinal cord by stimulating TAFA4+ C-LTMRs [95,98], and on the other, by increasing the secretion of oxytocin [99]. In humans, stimulation of CT afferents could be linked to increased tone of the parasympathetic system [100] and to a modulation of the activity of the endogenous μ-opioid receptor system [101]. More recently, a model of pain modulation has been hypothesized in which CT afferents would act by inhibiting ascending nociceptive pathways (bottom-up inhibition) and downregulating brain regions such as the anterior cingulate cortex and the insular cortex, two strategic cortices implicated in pain-experience perception [102].

It has, in fact, been demonstrated that the activation of CT afferents is able to reduce the temporal summation of a second pain, a neurophysiological marker of central sensitization and the pathophysiological basis of some CP syndromes like FMS [103]. The mechanism of action by which the stimulation of CT afferents would block the wind-up phenomenon in case of repetitive noxious heat stimulation would be based on the activation of inhibitory neurons which, in turn, would inhibit C nociceptors at the substantia gelatinosa-level of DRG [103]. Certainly, in healthy controls and acute pain studies, this model might have its own validity, but in CP patients where the maladaptive plasticity occurs, it could be less applicable and suitable. In fact, during the process of pain becoming chronic, there is a progressive shift in activity from the somatosensory cortices that process nociceptive inputs to the brain regions that regulate the emotional–affective components of the painful experience [104]. At the same time, it has been highlighted that to obtain pain modulation from the stimulation of the “affective system C”, obviously, the CT afferents must be intact [105]. In this prospective functional magnetic-resonance imaging and psychophysical study, the authors demonstrated that while in healthy subjects the activation of CT afferents is able to modulate heat pain regardless of its applied intensity, in patients with small-fiber neuropathy this clinical response does not occur, indirectly confirming IENFD loss in this clinical sample. In addition to this, the authors concluded by arguing that CT fibers could exert their pain modulation at the DRG level in lamina I-II by activating inhibitory interneurons [105]. More recently, the same research group explored the analgesic properties of CT afferents in a sample of patients with complex regional pain syndrome (CRPS) [106]. Although they observed a reduced threshold of heat sensation after repetitive stimulation of CT fibers and regardless of the presence of allodynia, the overall intensity of pain did not decrease, suggesting that the analgesic power of the “tactile affective system” is still present in patients with CP, even if too weak to have a significant effect [106]. On the other hand, a widespread reduction in IENFD and perhaps CT fibers was found in patients with CRPS, which explains to some extent the results of this study [107]. Unlike the results obtained in patients with chronic CRPS, a study conducted on a sample of different CP conditions found a significant and rapid improvement in pain intensity after an adequate CT-fiber stimulation paradigm (23% of pain reduction after only 11 min of CT-fiber stimulation at 3 cm/s with a force of 2.5 mN) [108]. Interestingly, in the same study, the authors found a lower accuracy and confidence in the interoceptive processes of their own internal body sensation in patients with CP, despite not being aware of it, supporting what is also reported in the scientific literature [109]. In this view, it has been highlighted that the relationship between pain modulation and body representation is bidirectional and reciprocal, with a dysfunction of one potentially predisposing the patient to the pathogenesis of the other, and vice versa. A potential therapeutic role of gentle, pleasant touch in modulating hypersensitivity of internal body sensation and associated elevated pain perception was demonstrated in an experimental study conducted on healthy controls [103].

Overall, the affective tactile system, represents an ancient phylogenetic system that not only has a fundamental function in the regulation of social behavior, but also has a peculiar role in the modulation of pain with both anti-nociceptive and pro-nociceptive properties [95]. Taken together, the above reported results are in line with the possibility that at least in CP conditions associated with C nociceptor dysfunction, stimulation of CT fibers may represent an alternative analgesic therapeutic approach [96,103,108]. From this perspective, an in-depth knowledge of the affective tactile system is crucial.

### 2.7. Attachment System (AS), C-Tactile (CT) Fibers, Oxytocinergic System (OS) and Brain Development

The largest organ of our body, the skin, is the first sensory channel of communication with the external environment, providing us with the sense of touch. It has been argued that social touch plays a crucial role in the development of a secure or insecure attachment. The observation of the high prevalence of insecure attachment in the psychiatric population, and the different pattern of perception of the stimulation of the C-tactile fibers, could represent the clinical aspects of this pathophysiological assumption [110].

In fact, affective touch and the related C-tactile (CT) fibers represent a fundamental functional system for the development of the newborn’s brain and, according to some evidence, also for that of the fetus [111,112,113]. It is known that the somatosensory system is almost completely mature by the 32nd week of pregnancy, with the exception of the insula and the superior temporal sulcus, two cortical regions involved in social cognition that become operational at around 2–3 months and 12 months of age, respectively [113]. It is interesting to note that it has been hypothesized that, already in this gestational period, the CT fibers may have a regulatory function on the maturation and differentiation of the “affective tactile system”. The rhythmic movement of the amniotic fluid in utero would stimulate the CT fibers, precisely through the delicate movements exerted on the “lanugo” of the fetus in a way similar to a physical massage [114]. Indeed, the effects of the therapeutic application of skin-to-skin contact and touch with both “kangaroo therapy” and pediatric massage have been studied in animal models and in premature human infants, demonstrating that CT-fiber stimulation improves growth, regulate the stress response, reinforces and modulates the immune system, and boosts the development of cognition and sensory–motor integration in the preterm human infant [115,116].

It is also important to highlight the fact that the maturation of the CT afferents means that they are already ready to carry out their function in the last gestational trimester, preparing the increase in the parasympathetic tone of newborns by reducing the “child’s stress for the birth” [117,118] and in preterm infants [119], and the analgesic power of the caresses provided by the parent–infant bond [120].

After birth, it was found that affective touch continues to exert a regulatory function on the development of the somatosensory system, bodily stress system, and autonomic nervous system, as well as the child’s immune system [121]. Overall, it has been documented that affective touch by an attachment figure (parent or caregiver in general) is able to promote the global development of the child by stimulating five different but interconnected systems: the somatosensory system, the autonomic system and immunity, the affiliative bond and social cognition [121].

From an evolutionary point of view, the CT-fiber system of the higher primate may have modified its original functions, moving from a role as a simple regulator of the stress response to a sort of homeostatic system responsible for controlling cognitive and emotional functions [111,112].

One of the most important mediators that parent–child bonding, affective touch and CT-afferent stimulation use to promote social affiliative bonds and prosocial behavior is oxytocin (OT), a neuropeptide composed of just nine amino acids synthesized by the hypothalamic paraventricular nucleus (PVN) and supraoptic nucleus. The OT receptors are widely express in the brain and in the entire organism, underlying the importance of this neuropeptide in regulating several functions in the body. In rodents, the main brain source of OT has been shown to be magnocellular neurons and, to a lesser extent, parvocellular neurons of the PVN and supraoptic nucleus. These neurons, with their axons and dendrites, reach different brain regions such as the frontal cortex, cingulate cortex and insular cortex, the basal ganglia, the limbic system (amygdala, hippocampus and septum), the midbrain, the brainstem and the spinal cord [122]. Furthermore, some parvocellular neurons also project to the anterior pituitary, where they regulate the release of adrenocorticotropic stress hormone [123]. Indeed, oxytocinergic projections from the PVN to the brainstem and hypothalamic–pituitary–adrenal axis have been considered the anatomical and functional pathways responsible for the anti-stress and analgesic action of OT [121,123,124,125]. In fact, the increase in parasympathetic tone due to the activation of CT fibers recorded in the fetus during the last three months of pregnancy could be mediated by the stimulation of the nucleus of the solitary tract of the brainstem by the oxytocinergic pathways coming from the PVN [126].

Importantly, it has been suggested that the first activation of the oxytocinergic system (OS) depends on the projections of CT fibers that reach the PVN from the insular cortex [127], supporting the role of affective touch in the development of CT fibers and the OT system and, more generally, for the regulation of pain modulation and the reward system.

Related to this, it has been demonstrated that OT acts synergistically with the dopaminergic afferents of the mesocorticolimbic regions and with the opioidergic tachykinin + neurons of the reward system and, more specifically, with those of the lateral and ventrolateral periaqueductal gray matter [113,121,128,129].

Taken together, the data reported above suggest that a normal development of CT fibers is fundamental for the physiological activation of the OS and the reward systems (dopamine and opioids) and at the same time for the pain modulation system. From this perspective, it has been found that, in newborns, the experience of pain is very similar to that of adults and that the caregiving environment is able to modify newborns’ sensitivity to pain [128]. In particular, skin-to-skin contact with caregivers has proven to be a valid non-pharmacological treatment for relieving pain in newborns in intensive care [130].

It is important to underline that it has been shown that from childhood to adolescence, pain represents a risk factor for the development of a dysfunctional attachment model, while the caregiving environment can constitute a predisposing or protective factor [131]. A schematic representation of the above data is shown in Figure 3.

### 2.8. Role of Oxytocinergic System (OS) in Attachment System (AS) and Pain Modulation

Oxytocin (OT) is a very ancient nonapeptide dating back almost 500 million years, initially conceived as a system for the regulation of basic functions, such as thermoregulation and energy balance [54]. Interestingly, over millions of years, OT analogue genes and related peptides have been highly conserved, but at the same time have undergone enormous evolutionary transformations structurally and functionally in order to adapt the organisms to rapidly changing environments [132,133]. The importance of OT, throughout evolution, is suggested by its multiple social and non-social functions common to different species, ranging from nematodes to humans [133]. In particular, in mammals, OT has been incorporated into labor and breastfeeding and then into the context of the mother–infant dyad, playing a critical role in the maturation process of the infant’s stress-reduction system [54,132,133]. Indeed, the oxytocinergic system (OS) in both zebrafish and mammals is part of the hypothalamic paraventricular “stress response” network, which, together with the locus coeruleus, the area postrema and the caudal hypothalamus, modulates the defense behavior in cases of noxious and other threatening stimuli [134]. In other words, the OS represents the biological system that allows individuals to respond in an adaptive way to various life adversities, regulating the psycho–immune–endocrine axis and promoting resilience [54]. According to this model, the OS, together with the affiliative brain and bio-behavioral synchrony represents a fundamental pillar of the neurobiology of affiliation and resilience [54]. From this perspective, it has been shown that the infant’s OS, during the sensitive period (the first nine months of life) and throughout childhood, is hardwired and programmed through the parental behavior and care (both mother and father) in a sort of “rhythm of safety”, supporting the importance of secure attachment (SA) for social, emotional and cognitive development of the child’s brain [54]. In practice, the OS is the physiological system that regulates the bio-behavioral synchrony of the mother-child dyad. This external regulation of an immature brain from a mature brain is key to the programming of cognitive and emotional functions such as executive functions, empathy, stress management, behavioral adaptation, conflict dialogue and behavior health, which are of crucial importance for social behavior and which characterize the resilient brain [54,135].

It has been demonstrated that OT can exert an anti-nociceptive action, together with anti-inflammatory and anxiolytic effects, thus improving perception and behavioral response to painful or aversive stimuli [136,137,138,139]. In particular, OT can exert its analgesic effect at both spinal-cord level and above-spinal-cord level, as well as in the peripheral nervous system. By interacting synergistically with the endogenous opioid system and activating γ-aminobutyric acid inhibitory interneurons, OT has been shown to inhibit nociceptive afferents (C and Aδ fibers) at the level of the most superficial layers of the dorsal horns of the spinal cord [136,139]. However, although the precise mechanism is not yet fully clarified, numerous data suggest that OT could be a modulator of pain perception by acting in several brain regions implicated in the cognitive–affective modulation of the pain experience [139]. In particular, the widespread presence of OT receptors in brain regions such as prefrontal cortex, orbitofrontal cortex, insular cortex, amygdala, and anterior cingulate cortex, supports the modulatory action of OS on the “medial pathways of suffering” of pain [139,140]. Furthermore, it has been hypothesized that OT could enhance the descending inhibitory system by stimulating the opioid system at the periaqueductal gray matter and at the rostral ventromedial reticular structure, two other brain regions with extensive expression of OT receptors [139]. Nonetheless, OT, by strengthening the reward system on the ventral tegmental area and the dopaminergic nucleus accumbens and reducing fear behavior by inhibiting the central nucleus of the amygdala, could protect against the chronicity of the painful state.

Interestingly, good functionality of both the reward system and the opioid system provided by the OS is the clear hallmark of an SA, further supporting the mutual connections between SA, OS and pain modulation [54,139].

A recent study conducted on a sample with chronic pain (CP) suggested that the analgesic effect of mindfulness-based pain management could be linked to the action of OT by reducing pain sensitivity and improving mood symptoms. Interestingly, a trend towards a reduction of inflammatory markers (IL-1b, IL-6, IL-8, and TNF-a) and stress biomarker dehydroepiandrosterone-sulfate was also observed in this clinical study, arguing that the anti-stress and anti-inflammatory properties of mindfulness-based pain management could be mediated by OT [141].

On the other hand, during the different phases of life, particularly in the first ones (e.g., the sensitive period of the first nine months), many variables, including type of attachment, sex, social experience, genetic and epigenetic factors, could have a negative impact on the physiology of the OS, which, in turn, over the course of life, could be less efficient in regulating the stress response, allostatic load/overload and resilience, leading to different outcomes [124].

Interestingly, a spectrum of fibromyalgia-like clinical symptoms has been shown to exist in the general population, within a continuum of biopsychosocial distress, leading to the concept of “fibromyalgianess” and the label of FMS [142,143,144].

### 2.9. Attachment System (AS), C-Tactile (CT) Fibers, and Oxytocinergic System (OS) Role in Other Central Sensitivity Syndromes (CSSs) and Chronic Pain (CP) Conditions

It has previously been reported that, while a secure attachment constitutes a protective factor against the pathophysiological development of chronic pain (CP), an insecure attachment, especially in its anxious form, could, on the contrary, represent a predisposing factor for the onset of CP.

In the field of CSS research, few data are available on the study of the attachment system (AS), the C-tactile (CT) system and the oxytocinergic system (OS). In this regard, in addiction to fibromyalgia, among the CSSs, aside from CSS fibromyalgia, migraine is undoubtedly the most studied painful condition. In this chronic disorder, insecure attachment has been found to be linked with higher disability, even in patients with episodic migraine [145]. In fact, compared to the general sample, migraine patients, particularly those with more severe disease, would be more frequently affected by an insecure attachment associated with the perception of having less social support [146]. Interestingly, an association between childhood trauma, insecure attachment, and migraine has also been documented [147], supporting our general model that, across CSSs, early stressful life events, along with an insecure attachment, could predispose the patient to central sensitization pathogenesis. On the other hand, it has been shown that an ambivalent attachment style could represent an important risk factor for pain severity, anxiety, depression and somatization in children with migraine [148], supporting the fact that the pathophysiological link between a dysregulation of the attachment system and migraine may begin in the early stages of brain development.

Interestingly, altered function of CT afferents was found in migraine patients during the interictal phase, during which increased habituation to stimulation of CT afferents on trigeminal-innervated skin was shown [149]. The authors concluded that the reduction in pleasant tactile experience and increased lack of habituation to stimulation of CT afferents observed in migraine patients could be secondary to subclinical interictal mechanical allodynia [149]. Regarding the oxytocinergic system, it has been shown that mean salivary [150] and plasma levels [151] of oxytocin are higher in patients with chronic migraine than in healthy subjects. Interestingly, this finding goes in the opposite direction to other chronic pain conditions where the oxytocin levels have been found to be lower in patients compared to healthy controls [152].

Among the CSSs, we have already highlighted the importance that post-traumatic stress disorders (PTSD) have in the clinical evaluation of FMS [47] and we have already underlined the fact that having an insecure attachment predisposes one to a greater risk of developing a psychiatric disorder later in life [110]. In fact, as also supported by our group, it has been hypothesized that dysfunctional somatic sensory integration could be the missing link capable of explaining the high correlation between early traumatization, insecure attachment and the subsequent development of psychopathology and PTSD [110,153]. In particular, in PTSD, hyper-reactivity to low-threshold stimuli could depend on the persistent high state of arousal recorded in these patients, which, in turn, could be responsible for the perception of harmless stimuli as threatening [153,154]. Interestingly, individuals with PTSD report a low pleasantness or even an unpleasant and intense response to tactile C-fiber stimulation, suggesting a potential role of this class of C-fibers in determining the general hyper-responsiveness to stimuli observed in PTSD [155,156,157]. From this point of view, it is interesting to note that it has been discovered that subjects who have experienced a lack of emotional contact together with experiences of abandonment or abuse in the initial stages are characterized by an attenuation of social contact [158] and lower plasma levels of oxytocin [159,160]. These findings were considered indirect evidence of the role of disruption of the CT system in the genesis of insecure attachment and in the pathophysiology of the oxytocinergic system, also in PTSD [153]. Indeed, low salivary levels of oxytocin have been found in police officers suffering from PTSD, supporting a dysfunction of the oxytocinergic system in this psychiatric disorder [161].

Another interesting field of research to report in this session is represented by the relationship between the attachment system and irritable bowel syndrome (IBS), another central sensitivity syndrome frequently observed in clinical practice. In fact, visceral hypersensitivity and a low pain threshold represent the distinctive features of IBS, a disorder characterized by extra-intestinal symptoms such as cutaneous hyperalgesia and widespread pain, which suggest a central dysfunction of the nociceptive process even in this chronic disorder [162,163].

A high prevalence of insecure attachment has been reported in patients with IBS, with the anxious type being even more evident than that seen in inflammatory bowel disorder [164]. This behavioral trait, particularly its fearful type, has been consistently found in different cultural and geographic sites, and would have a negative impact on IBS by increasing catastrophizing and pain-related beliefs [165]. According to other findings, it has been hypothesized that insecure avoidant attachment associated with a tendency towards somatization could play an important role in the development of IBS [166]. From this perspective, together with insecure attachment, the positive childhood history of recurrent abdominal pain observed in adult patients with IBS could support the progressive nature of the disease starting from the early stages of life [166]. In patients with IBS, the prevalence of insecure attachment would be even higher than that observed in patients with panic disorder, and would depend on the presence of a dysfunctional paternal bond, although this recent observation needs to be further confirmed [167].

Finally, although no conclusive considerations can be reached, an old study conducted on the analgesic power of oxytocin in patients suffering from IBS has shown that the effectiveness of this neuropeptide could depend on the inhibition of nociceptive afferents at the level of the dorsal horn of the spinal cord [168].

In clinical practice, another frequent type of central sensitivity syndrome is unspecific chronic low-back pain. A reduction in the pleasantness of touch was found in patients with low-back pain, to a lesser extent in both the subacute and chronic phases. The authors suggested that the dysfunction of the brain’s pleasant representation of touch mediated by CT-afferent abnormalities could be considered a marker of transition from the acute pain phase to a CP syndrome [169]. In line with this trend, a reduction in pleasantness through stimulation of CT afferents was also found in patients with post-herpetic neuralgia and complex regional pain syndrome (CRPS) [107]. Interestingly, in this study, the typical inverted-U curve of CT afferent stimulation (Table 2) was not found in patients with post-herpetic neuralgia, confirming the prevalent involvement of C-fiber damage over A-fiber damage in this CP condition [107,170]. Furthermore, in patients with unilateral CRPS, reduction in pleasantness due to stimulation of CT neurons was observed not only in the affected side, but also in the contralateral side, supporting a more systemic involvement of CT-fiber system dysfunction in CRPS [107]. On the other hand, as reported above, a reduction in intraepidermal nerve fiber density (IENFD) has been found in several chronic disorders, regardless of the presence of CP [84]. Therefore, the bilateral deficit in pleasantness from CT-afferent stimulation observed on CRPS is not surprising, since a bilateral reduction in IENFD was also observed in this CP disorder in its unilateral form [107,171].

In any case, these data suggest that a dysfunction of the CT-fiber system may be common to several CP conditions probably characterized by a subclinical reduction in the perception of pleasantness. This could represent the missing link capable of explaining the pathophysiological mechanisms shared by chronic overlapping pain conditions [96] although, despite the large amount of data available, it is still a subject of debate as to whether the attachment system plays a key role in the pathogenesis of CSSs and functional somatic disorders [172].

## 3. Discussion

One of the most recent and interesting debates on the pathophysiology of fibromyalgia syndrome (FMS) concerns the prevalent role of peripheral or central pathogenic mechanisms along nervous system pathways in combination with immune system dysregulation [36,173].

In their theoretical model called ‘Fibromyalgia: Imbalance of Threat and Soothing Systems’, Pinto et al. (2023) [36], define FMS as a disease linked to a dysfunction in emotional regulation that leads to an increase in the perception of threat and a reduction in the sense of safeness and protection. Specifically, the hypothetical ‘Fibromyalgia: Imbalance of Threat and Soothing Systems’ model would have been based on three main and interconnected pathophysiological factors: (1) a high perception of threat; (2) a reduced activity of the soothing affiliative system; (3) a persistent activation of the brain’s salience network [36]. This hypothesis found support in a recent study which highlighted the presence of emotional dysregulation with greater arousal towards unpleasant and socially unpleasant images in patients with FMS [174].

The ‘Fibromyalgia: Imbalance of Threat and Soothing Systems’ theory has also been questioned because it would not be a truly new theory but rather a different way of representing the model of autonomic nervous system (ANS) dysregulation. Furthermore, in this model, there is no mention of the role of small fibers and the immune system activation on the dorsal roots of the spinal ganglia which have recently been considered to be potentially involved in the pathophysiology of FMS [87,173].

In our opinion, there is no real controversy between the two points of view, but rather a different perspective from which the authors see the phenomenon.

Certainly, it is still a matter of debate whether chronic painful states can be the result of “bottom-up” amplification mechanisms of nociceptive afferents or whether, on the contrary, a dysfunction of the descending nociceptive inhibitory system can lead the increase in pain perception towards a chronic state.

A different and more complete interpretation could be proposed by returning to clinical practice and the clinical phenotype of FMS and, more generally, of all central sensitivity syndromes (CSSs). There is no doubt that FMS should not be simplistically considered a chronic pain (CP) syndrome, both from a clinical and pathophysiological point of view. In clinical practice, in fact, CSSs are characterized not only by pain but, above all, by other conditions such as sleep disorders, cognitive problems, fatigue and psychiatric illnesses (depression, post-traumatic stress disorder, and panic disorder), which are often even more disabling than the pain itself and which are heterogeneously reported by patients.

As already widely reported above, despite the contradictory results probably linked to the complexity of human-stress neural circuits and other factors already discussed, several scientific results and data support the theory of FMS as a stress-related disorder. In fact, a large body of evidence on hypothalamic–pituitary–adrenal (HPA) axis and ANS dysregulations has been found in FMS and more generally in other centralized pain syndromes [17]. In this vision, FMS and other CSSs can be considered a group of diseases characterized by hypersensitivity to the perception of pain and, more specifically, to the anomalous interpretation of the “threat” of innocuous sensory inputs, both internal and external, to the body [17,36].

It remains to be understood how this deficit in the interpretation of the salience of stimuli and, in particular, in the discrimination between threat and safety occurs in FMS and probably in other CSSs. According to extensive scientific evidence from both animal and human studies, we believe that, at least to a certain extent, the oxytocinergic system (OS) can represent a coherent answer to this question [175]. It has in fact been hypothesized that oxytocin (OT) may play a crucial role in discriminating threat stimuli from safety stimuli, being a neuropeptide that is part of the hypothalamic paraventricular stress-response network, widely represented in nature, from zebrafish to mammals [134,175]. In particular, it has been shown that OT is specifically involved in the recognition of the threat that could come from the environment or from fearful facial expressions. Specifically, the role of OT would be to identify a potential source of danger and consequently activate adequate adaptive responses [175]. The broad distribution of OT receptors within brain regions involved in stress regulation such as the prefrontal cortex, limbic area, hypothalamus, raphe, and medulla oblongata is in line with the OS functions mentioned above [176].

On the other hand, the analgesic power of OT has been widely attributed to its ability to improve the behavioral response to painful and aversive stimuli by reducing the perception of negative salience [139]. Indeed, it is widely demonstrated that OT receptors are widespread in different brain regions of the so-called “medial pain pathways” [139,140].

As previously described, it is important to underline the fact that the analgesic action of OT is not limited to the modulation of the perception of the painful experience at the level of the higher brain centers, but extends to the spinal cord by activating the descending inhibitory system effect and exerting a direct inhibitory effect on nociceptive afferents at the dorsal horn level [136,139]. It is also hypothesized that the physiology of OS, by strengthening the reward system (ventral tegmental area and nucleus accumbens) and inhibiting the central nucleus of the amygdala [139], could represent a crucial mechanism of protection from the transformation of pain from acute to chronic.

Finally, in animal models, a large body of evidence supports the systemic anti-inflammatory properties of OT physiology. For this reason, the role of OT could be fundamental for preventing the pathogenesis of inflammatory pain, neuropsychiatric disorders and other chronic-degenerative diseases considered linked to systemic inflammation, such as cardiovascular and gastrointestinal diseases, diabetes, and obesity [177]. On the other hand, it has been highlighted that OS might be susceptible to inflammation from early life and that neonatal inflammatory pain may lead to its dysregulation and to neurodevelopmental diseases in later life [178].

In fact, it is important to underline the fact that the OS is a very dynamic biological system and that it is not completely mature at birth, as it is subjected to intense programming during the first nine months of age and throughout childhood through the maternal brain and caregiving behavior [54]. As already reported in the previous paragraphs, OS has a key role in the regulation of bio-behavioral synchrony of the mother–child dyad. For this reason, a mature maternal brain with a physiological OS has a key role in the physiological development of the immature infantile brain and consequently in the executive cognitive functions, emotional regulation functions, empathy, stress regulation, behavioral adaptation and more generally the development of the resilient brain [54,135]. In fact, a resilient person is characterized by lower reactivity of the HPA axis and a higher parasympathetic tone in response to stressful conditions, compared to a subject without a resilience trait [124]. On the other hand, a large number of scientific findings support the detrimental effects of adverse events in early life on OS, with long-lasting negative consequences on both the physiological and behavioral development of the child [179]. From this perspective, we shed new light on the epidemiological data and the pathophysiological role of early adverse events and, more generally, of stressful events in the pathogenesis of FMS and comorbidity with post-traumatic stress disorder [39,46,47,48,49].

As previously described, it has been reported that insecure attachment (IA), the behavioral analogue of an OS dysfunction, predisposes one in adolescence to emotional dysregulation and an increase in the response of the stress system secondary to stress factors related to attachment [58]. In adulthood, this behavioral pattern has been found to be associated with biological, psychological and sociological dysfunctions, highlighting the importance of functional programming of the OS and the consequent secure attachment (SA) in the initial phase of brain development [61]. In other words, a dysfunction of OS and IA can be considered as two sides of the same coin, which, in a single construct, represent an important risk factor for several adult diseases including CP [61,63,64,65,66,67].

In our model, the pathophysiology of FMS would be a progressive and dynamic process that would have its “primum movens” in CT-fiber-system abnormalities and affective touch disruption.

In fact, the CT-fiber system represents the key physiological channel in the regulation of pleasant sensations, having a synergistic action with OS and the opioid system [101,139]. This system is ready to act as a regulatory program for the maturation and differentiation of the “affective tactile system”, already during the last trimester of pregnancy [111,112,113]. On the other hand, there is the importance of having an SA to achieve good functionality of the reward system and the opioid system, supporting the functional connections between SA, OS and pain modulation [54,139]. Nonetheless, a CT-fiber dysfunction linked to opioid system abnormalities [93], together with a dysregulation of insular circuits [94], was observed in patients with FMS. These findings support the hypothesis that a widespread subclinical dysfunction of the CT-fiber system could be a common pathophysiological ground that predisposes one to various CSSs, characterized by a subclinical reduction in the perception of pleasantness, regardless of the presence of pain [96,107]. This could shed light on other otherwise incomprehensible prevalence data, relating to the reduction of IENFD in various diseases and even in pain-free clinical samples [84].

From this perspective, it has been demonstrated in animal models that a sustained increase in bilateral glutamatergic activity of the insular cortex, the first cortical station for CT afferents, is responsible not only for the pathogenesis of pain but also for the loss of peripheral fibers. According to their findings, the authors hypothesized that the high prevalence of small-fiber pathology and IENFD observed in FMS might be mainly due to insular cortex dysfunction [86]. Interestingly, psychophysiological studies have shown that the skin stimulation of C-low-threshold mechanoreceptor afferents with stroking at intermediate frequency speeds performed in healthy human controls subjected to tonic muscle pain has been shown to be capable of producing a type of allodynia very similar to that observed in patients with widespread pain, such as FMS [95,180,181].

By putting together all the pieces of this scientific puzzle discussed through the different topics addressed in this overview, we hypothesize that our pathophysiological model could be applicable to FMS, and CSSs, and probably to all chronic-degenerative diseases.

In nature, there is an ancient phylogenetic and physiological system functional for the survival of life, common to many living species but species-specific, composed of at least three subsystems: the attachment system, the affective CT-fiber system and the OS. Over the course of millions of years, this multisystem physiological apparatus, particularly in the human race, has undergone a modification from its original functions as a simple stress regulator, towards a homeostatic system. This homeostatic system would be responsible for the emotional and cognitive regulation of social stimuli and, in particular, for differentiating what could be a real threat while maintaining the tendency to remain in a safe situation (e.g., affiliation system) [111,112,132,133]. With its neurological, psychological, immunological and endocrine subsystems, this integrated system begins to be programmed from the last three months of pregnancy until childhood, through bio-behavioral synchrony, a biological and physiological function that is elaborated by the parent–child dyad [54,135]. The epigenetic characters of this system make it dependent on the type of attachment system, from the first stressful life events, in particular from the last period of pregnancy and from the first nine months of life (e.g., sensitive period) up to childhood and early adolescence. All these crucial modifiable risk factors of neurodevelopment may be able to influence the neurobiological path and development of the infantile brain [178,179]. The final result of the consequent maladaptive brain, could lead progressively to a dysregulation of the bodily stress system together with cognitive–emotional dysfunction, systemic inflammation, central sensitization, and an increasing sensitivity and perception of pain.

On the other hand, the clinical heterogeneity of syndromes such as FMS, CSSs and functional somatic disorders could also be linked to the type of attachment of the patients, to sex, to genetic and epigenetic factors, and, particularly, to the life stage of the patients at the time of clinical observation [124,179]. In line with this, it has been found that in the general population a spectrum of fibromyalgia-like clinical symptoms exists within a continuum of biopsychological distress, which has been termed “fibromyalgia” [142,143,144].

### Therapeutic Implications

As widely reported, a large amount of data would support the use of oxytocin (OT) as a pain modulator in clinical practice, particularly in chronic pain (CP) conditions with deep pressure pain such as fibromyalgia syndrome (FMS) and in patients with back pain, abdominal pain, and migraine [152,182]. The few available studies obtained on human samples have shown inconclusive results and certainly, the paucity of studies, the heterogeneity of clinical samples, the different design of the studies, the mode of administration, the dosage and the pharmacokinetics of OT, are among the most important reasons for these conflicting results [152].

To the best of our knowledge, the only published clinical study on the effectiveness of intranasal OT in reducing pain in patients with fibromyalgia showed negative results, but, as also stated by the authors themselves, this study presents many limitations that do not allow conclusions to be drawn definitively [183]. Indeed, a better understanding of the OT receptor system will help explain the mixed results of exogenous OT applications in humans [175], keeping in mind that the effects of OT strongly depend on gender, social context, early life experiences, route of administration, and epigenetic mechanisms, all variables that will need to be taken into consideration for future studies [175,179].

As already cited, depending on the different phases of life (e.g., the first nine months of life, a sensitive period), various risk factors (type of attachment of parent in the parent–infant dyad, type of attachment of patient, social experience, genetic and epigenetic factors), could have a negative impact on the physiology of the oxytocinergic system [54,179] which in turn, could have a different type of response to exogeneous administration of OT.

The analgesic efficacy of gentle touch due to the activation of C-tactile (CT) fibers has been explored, both in healthy subjects for acutely induced pain and in patients with CP, showing a potential therapeutic role of this non-pharmacological treatment in CP conditions [96,97,103,106]. A study conducted on several CP syndromes supports the role of interoceptive tactile stimulation as a non-pharmacological and complementary approach for the treatment of CP [100]. However, in future studies, it will be important to explore the therapeutic potential of CT-fiber stimulation also in chronic patients with small-fiber pathology, which represents a common condition in patients with FMS and many other diseases [83,84].

A significant clinical approach is that of the neonatal intensive care unit where, in the preterm newborn, a family education intervention, according to the calming cycle theory, is able to accelerate the maturation of the preterm newborn’s vagal control, highlighting the importance of the stimulation of the CT fibers and other sensory components of the bio-behavioral synchrony [184]. In this clinical setting, a nurture specialist represents “a human tool” capable of bidirectionally promoting the system of social involvement between mother and infant [184]. From this perspective, in clinical practice it is important to explore the type of attachment of patients with CP, together with that of the caregiver, in particular in patients with FMS who usually do not have good adherence to pharmacological therapy. In particular, it appears essential to evaluate the type of relationship that is established in the caregiver–patient dyad and to encourage the patient’s social support by making the caregiver participate, where possible, in the therapeutic project. We strongly believe in the therapeutic power of the psycho-educational part of the visit, capable of strengthening the compassionate physician–patient relationship, and promoting the social system of involvement within the physician–patient dyad.

On the other hand, the biopsychosocial model for the clinical evaluation and treatment of patients with CP and central sensitivity syndromes is based on a patient-centered approach in which individualized and multidisciplinary treatments should always be specifically tailored to each patient and set based on life history and clinical symptoms [17,185].

We strongly believe that the scientific community should consider extending research in the field of pathophysiology of chronic overlapping pain conditions, functional somatic disorders, and central sensitivity syndromes, by evaluating the role of the attachment system, CT-fiber system and oxytocinergic system, in a multidisciplinary context.

Improving knowledge of the pathophysiological mechanisms of these functionally integrated systems, probably crucial not only in CP conditions but also in the pathogenesis of chronic-degenerative diseases, could provide doctors and patients with valuable clinical tools to improve the psychological and social functioning of subjects affected by these chronic disorders.

Even more important could be the positive impact on prevention through scientific knowledge-based awareness campaigns for healthcare professionals, patients and the general population.

A reduction in the burden of the disease at an individual and social level, and, above all, a promotion of cognitive neuroscience and social sciences, should always be pursued in a civil society that looks to the future.

## 4. Conclusions

We argue that the crucial pathophysiological mechanism responsible for fibromyalgia and other central sensitivity syndromes is the dysfunctional development of the bodily stress system.

We hypothesize that the attachment system, the C-tactile fibers and the oxytocinergic system are the three strategic and potentially modifiable risk factors responsible for the dynamic process that leads to dysregulation of the stress response and its key neurobiological factors: hypothalamic–pituitary–adrenal axis, sympathetic-adrenal–medullary, and insular–amygdaloid circuit. From this point of view, insecure attachment, in a synergistic relationship with the stressful events of childhood, could represent the “primum movens” capable of sensitizing the threat system during the “sensitive period” of brain development. In this neurobiological process, the anomalous stimulation of the C-tactile fiber system and, consequently, the epigenetic modification of oxytocinergic system (oxytocin and related receptors) by dysfunctional parental care during the early stages of life, could initiate central sensitization through alterations of the stress-response system.

Over time, these mechanisms could lead to dysregulations of subcortical and cortical brain networks (e.g., insula, amygdala, cingulate cortex, dopaminergic and opioid system (reward system), nucleus of the solitary tract, and prefrontal cortex), physiologically involved in pain modulation, emotional regulation, immune system activation and in the selection of adaptive behaviors.

In adulthood, a vicious circle characterized by a progressive reduction in resilience with respect to stressful life events and increased activation of a threat system could represent the “natural” biopsychosocial trajectory of this dysfunctional psychological, neurological, endocrinological, immunological process.

The different clinical phenotype within and between central sensitivity syndromes might depend on biological (epigenetic), psychological and social factors expressed individually by each patient and his or her history. In clinical practice and increasingly in scientific trials, it is essential to explore in depth the patient’s real situation and his stage of the disease, while the diagnostic category must be replaced by the clinical dimension, which must be considered not only a static concept but rather a constantly evolving dynamic process.

The label “fibromyalgianess” and its meaning are representative of this pathophysiological and clinical model in which peripheral and central hypersensitivity and multisymptomatic syndromes cannot be considered as single pathological entities but rather as different clinical manifestations with the same primordial pathophysiological substrate (graphical abstract).

## Figures and Tables

**Figure 1 biomedicines-12-01744-f001:**
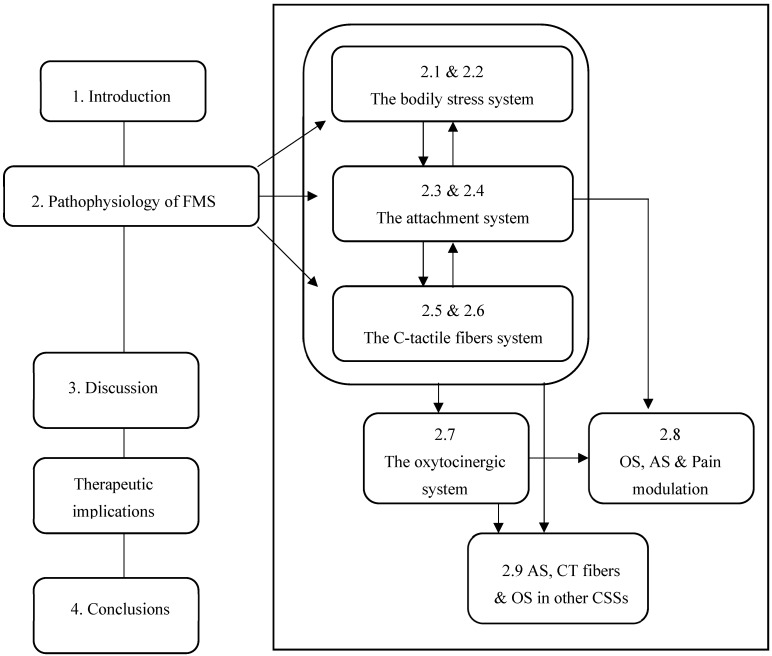
Flowchart of logical and biological connection of the topics covered in the narrative review.

**Figure 2 biomedicines-12-01744-f002:**
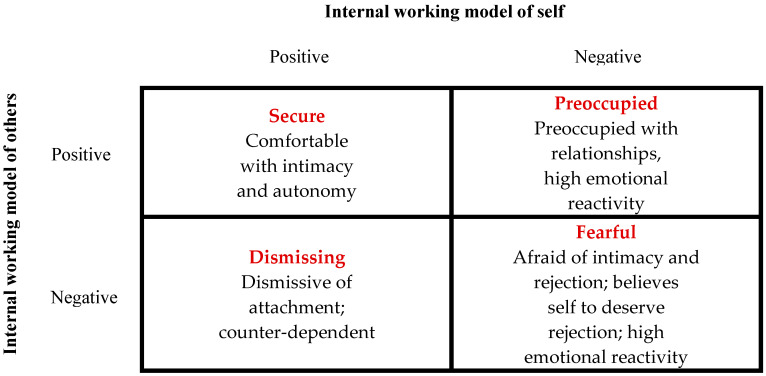
The four subtypes of adult attachment style from Ma (2006) [55], according to Bartholomew and Horowitz’s model (1991) [56].

**Figure 3 biomedicines-12-01744-f003:**
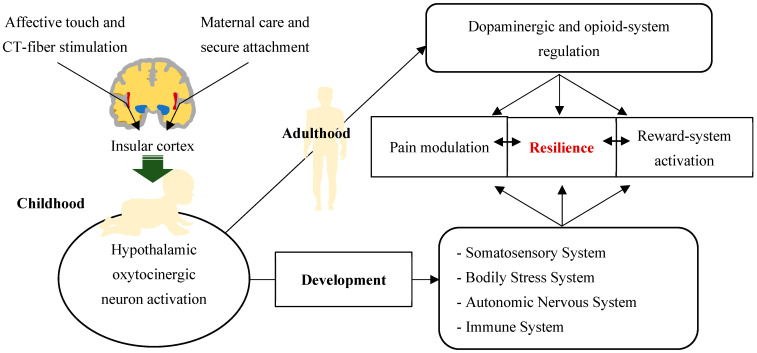
Good resilience, reward-system activity and pain modulation as a result of a physiological functional integration between the attachment system, CT-fiber system, and oxytocinergic system. Maternal care and the development of a secure attachment with the stimulation of CT fibers through affective touch are able to activate hypothalamic oxytocinergic neurons via the insular cortex. The latter, during childhood, are fundamental for the physiological development of the somatosensory system, the bodily stress system, the autonomic nervous system and the immune system. During adulthood, the oxytocinergic system is also essential for the regulation of the reward system and the modulation of pain. These neurobiological networks promote adaptive behavior and resilience in both adulthood and childhood.

**Table 1 biomedicines-12-01744-t001:** Cluster of symptoms and related organ-system classification for the diagnosis of FMS according to the American College of Rheumatology diagnostic criteria, adapted from ref. [1].

Organ System	Symptoms
The nervous system	depression, thinking or remembering problems, numbness/tingling, nervousness, seizures, hearing difficulties, headache, dizziness, insomnia, blurred vision, ringing in ears
The gastrointestinal system	pain/cramps in the abdomen, constipation, pain in the upper abdomen, nausea, diarrhea, loss of appetite, loss of/change in taste, vomiting, oral ulcers, heartburn
The immune system	rash, sun-sensitivity, easy bruising, dry eyes, fever, dry mouth, Raynaud’s phenomenon, itching, hives/welts
The musculoskeletal system	muscle pain, fatigue/tiredness, muscle weakness
The urinary system	bladder spasms, painful urination, frequent urination
The respiratory system	chest pain, shortness of breath, wheezing
The integumentary system	hair loss

**Table 2 biomedicines-12-01744-t002:** The neurophysiological properties and characteristics of CT fibers in response to different stimuli (adapted from Ackerley et al., 2022) [80].

Type of Stimulus	Properties of CT Fibers	Characteristics of CT Fibers
**Mechanical**	Low mechanical-activation threshold	Monofilament activation threshold of <5 mN
	Sensitive to moving touch	Very responsive to different types of mechanical stimuli such as hand, brush, cotton wool, needle, smooth metal plate
	Responsive to both blunt and sharp probes	Equally responsive to sharp and blunt indentations
	Optimal activation with a stroking speed of 1–10 cm/s	Greater activation at intermediate velocities. Stroking velocity >10 cm/s and <1 cm/s produces lower activation (reversed U- shape response)
	Intermediate adaptation	Static application of the mechanical stimulus produces an initial burst of spikes followed within seconds by cessation of activation
	Firing after removal of a stimulus	Often a response of a few spikes is observed immediately after removal of the stimulus
	Reduction in activation with repeated stimulation	Tendency to significantly reduce activation with the repetition of stimulation
	Biphasic response to prolonged stimulus	Following prolonged stimulation, after a phase of cessation of activity, a second wave of activation is observed
	Vibration	Poor response to vibration with few spikes at 1 Hz of stimulation and transient response with a single phase-locked spike at vibrations between 16 and 50 Hz
	Skin stretch	Activation by stretching the skin with continuous weak discharge after prolonged, static stretch
**Thermal**	Radiant heating and cooling	Poorly responsive to thermal stimuli with low response rate to evaporative cooling
	Stationary mechano-thermal stimulation	Better reactivity to skin temperature compared to warm static touch
	Dynamic mechano-thermal stimulation	Better responsiveness to skin temperature than hot and cold touch, depending on the duration of stimulation
**Electrical**	Low-frequency, repetitive electrical stimulation, in conjunction with mechanical stimulation	In this type of stimulation paradigm, spike latencies are slowed slightly
	Activity-dependent slowing at 2 Hz stimulation	At this frequency stimulation, the elicited spikes have a very small latency delay
	High-frequency electrical stimulation	Short bursts of activation can follow electrical stimulation up to frequencies of 100 Hz, with increased response latency over 50 Hz

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
