# Peer review of "Insecure Attachment, Oxytocinergic System and C-Tactile Fibers: An Integrative and Translational Pathophysiological Model of Fibromyalgia and Central Sensitivity Syndromes"

_biomedicines, 2024, doi:10.3390/biomedicines12081744_

Round 1

Reviewer 1 Report

Comments and Suggestions for Authors

The manuscript "Insecure attachment, oxytocinergic system and C tactile fibers: an integrative and translational pathophysiological model of fibromyalgia and central sensitization syndromes" discusses the fundamental mechanisms of the pathophysiology of fibromyalgia conceptualized as stress intolerance syndrome by thoroughly describing  the potential role of the attachment system, C tactile fibers, and oxytocinergic system dysfunction in the pathophysiology of fibromyalgia syndrome and other central sensitivity syndromes. In addition, the article discusses about the therapeutic implications of this new global and translational pathophysiological model. It has been established by the finding of numerous clinical studies that fibromyalgia syndrome is generally characterized by chronic muskeloskeletal pain. Furthermore, C Tactile (CT) primary afferents are known to contribute to pleasant touch and provide an important sensory underpinning of social behavior. The submitted manuscript has succulently reviewed the relevant literatures to extract the main theories on the pathophysiology of fibromyalgia syndrome focusing on the role of the bodily stress system, thereby proposing a new pathophysiological model, common to different types of chronic pain syndromes and Central Sensitivity Syndromes (CSS).

After going through the manuscript, I have following comments for the authors.

1.     CSS represents a heterogeneous group of disorders such as fibromyalgia, irritable bowel syndrome (IBS), chronic headache, temporomandibular disorders (TMD), and pelvic pain syndromes. Various clinical studies have shown that these disorders share common symptoms, persistant pain being the prominent feature. Although, the main focus of the submitted manuscript is fibromyalgia syndrome, I would suggest the authors to briefly discuss the role of attatchment system, C tactile fibres, and oxytocinergic system dysfunction in the pathophysiology of other individual CSS disorders.

2.     Previously, it has been demonstrated that body reprogramming intervention is an effective approach for patients living with fibromyalgia and central sensitivity syndromes on a variety of clinical measures. I would suggest the authors to briefly discuss this therepeutic intervention in the manuscript.

3.     Figure 1 is inappropriately cropped, hence it needs to be improved.

4.     There are some one-sentence paragraphs in the manuscript (especially in Introduction section and concluding part). Please compile these one-sentence paragraphs together to make a bigger paragraph.

Comments on the Quality of English Language

The authors need to do a minor grammatical correction and some syntax adjustments in the manuscript.

Author Response

For review article

Response to Reviewer X Comments

1. Summary

I would like to thank you very much indeed for taking the time to review this not-so-short manuscript. Indeed, I really appreciated the advice I was given, recognizing its values. I have tried to answer your questions as best as possible. Please find the detailed responses below. You can find the corresponding revisions/corrections highlighted in red color in the re-submitted files. In the same files, you can also find the deleted sentences as well as the modified numbering and renaming of the paragraphs highlighted and crossed out in red.

3. Point-by-point response to Comments and Suggestions for Authors

Comment 1: [CSS represents a heterogeneous group of disorders such as fibromyalgia, irritable bowel syndrome (IBS), chronic headache, temporomandibular disorders (TMD), and pelvic pain syndromes. Various clinical studies have shown that these disorders share common symptoms, persistant pain being the prominent feature. Although, the main focus of the submitted manuscript is fibromyalgia syndrome, I would suggest the authors to briefly discuss the role of attatchment system, C tactile fibres, and oxytocinergic system dysfunction in the pathophysiology of other individual CSS disorders].

Response 1: [Thank you for this observation. We tried to satisfy your request as best I could, considering the lack of data published so far in this field. We added a subchapter (2.9) at page 25 (please see the added parts written and underlined in red in the file called “revised manuscript”)]

Comment 2: [Previously, it has been demonstrated that body reprogramming intervention is an effective approach for patients living with fibromyalgia and central sensitivity syndromes on a variety of clinical measures. I would suggest the authors to briefly discuss this therepeutic intervention in the manuscript].

Response 2:  [According to your useful suggestion, we have included a paragraph (page 8) dedicated to the model underlying the therapeutic approach, explaining the basic assumptions and making some comments referring to the results of the study by Lanario et al., 2023 (please see the added parts written and underlined in red in the file called “revised manuscript”)].

Comment 3: [Figure 1 is inappropriately cropped, hence it needs to be improved].

Response 3:  [Thank you very much for this point. We realized that this figure was not so clear in its meaning and for this reason we decided to change it completely making it more readable. Please, see the figure in the attachment file or directly in the revised manuscript. Note that now this figure is called number 2, page 13].

Comment 4: [There are some one-sentence paragraphs in the manuscript (especially in Introduction section and concluding part). Please compile these one-sentence paragraphs together to make a bigger paragraph].

Response 4:  [Following your suggestions, as best we could, we rewrote and reformulated some parts of the manuscript focusing on the introduction and final parts (please see the added parts written in red in the file called “revised manuscript”)].

4. Response to Comments on the Quality of English Language

Point 1: [There are some one-sentence paragraphs in the manuscript (especially in Introduction section and concluding part). Please compile these one-sentence paragraphs together to make a bigger paragraph].

Response:    (See response number 4)

5. Additional clarifications

[Here, mention any other clarifications you would like to provide to the journal editor/reviewer.]

Reviewer 2 Report

Comments and Suggestions for Authors

This paper, as stated in its title, presents an integrated and translational pathophysiological model of fibromyalgia and central sensitization syndromes. The authors aimed to develop a comprehensive biopsychosocial model of chronic pain syndromes. They provided many details in the 18 pages of text, however, the model was only presented in the graphical abstract. This paper lacks logical connections between facts that would support a solid biopsychosocial model. Instead, it appears to be a collection of facts and findings related to chronic pain and associated disorders.

I would like to make some remarks about the content of the paper.

The content of Figure 1 has to be improved for clarity.

The concept of "attachment" (secure and insecure) requires a more comprehensive introduction.

It is important to include information about the C-fiber system and its physiology.

Figure 2 presents a simple diagram that does not illustrate "physiological functional integration" supposed by the legend.

One more problem for readers is the huge amount of abbrevations. It is difficult to remember them all. Some of abbreviations are used only ones, such as International Association for the Study of Pain (IASP),: “Fibromyalgia: Imbalance of Threat and Soothing Systems” (FITSS); central sensitization (CS); prefrontal cortex (PFC); Orbitofrontal Cortex (OFC); insular cortex (IC); Rostral Ventromedial Reticular Structure (RVM) etc. Two times - ventral tegmental area (VTA); Nucleus Accumbens (NAc); periaqueductal gray matter (PAG) etc. Some abbreviations were not explained, such as DHEA-S.

The review would benefit from a more structured and coherent evaluation of the problem. In order to improve the clarity of this review, I suggest the following:

1. To better organize the content, a flowchart can be used. This review would benefit from following the guidelines of the Preferred Reporting Items for Systematic Reviews and Meta-Analyses (PRISMA) https://www.prisma-statement.org/.

2. To enhance clarity and conciseness, it is advisable to minimize the usage of abbreviations and acronyms. Please consider refraining from abbreviating the terms secure and insecure attachments.

3. The graphical abstract presents a conceptual overview that would benefit from more detailed explanations in the main text.

4. The text should be reorganized to improve logical flow and readability. Some sentences could be shortened to make the review more concise and clear. 

Author Response

For review article

Response to Reviewer X Comments

1. Summary

I would like to thank you very much indeed for taking the time to review this not-so-short manuscript. Indeed, I really appreciated the advice I was given, recognizing its values. I have tried to answer your questions as best as possible in order to make the document more readable and logically structured. Please find the detailed responses below. You can find the corresponding revisions/corrections highlighted and underlined in red color in the re-submitted file. In the same file, you can also find the modified numbering and renaming of the paragraphs highlighted in red to make the flow of concepts more logical.

3. Point-by-point response to Comments and Suggestions for Authors

Comment 1: [The content of Figure 1 has to be improved for clarity]

Response 1: [Thank you for highlighting this part of the manuscript. Indeed the figure 1 was not so easily understanble, especially without a more in-depth explanation of its content in the text. For these reasons, I decided to follow your suggestions not only by changing the scheme of figure 1, but also by explaining in more depth the attachment theory to which figure that now it has become figure 2 (page 13) (please see the added parts written and underlined in red in the revised manuscript and see also the answer to your comment number 2)]

Comment 2: [The concept of "attachment" (secure and insecure) requires a more comprehensive introduction].

Response 2:  [We appreciated your request especially because we recognize that a more detailed explanation of this topic made the manuscript clearer in its conceptual flow. We have decided to include a new subchapter specifically dedicated to this topic (page 11) (please see the added part written and underlined in red in the revised manuscript].

Discuss the changes made, providing the necessary explanation/clarification. Mention exactly where in the revised manuscript this change can be found – page number, paragraph, and line.]

“[updated text in the manuscript if necessary]”

Comment 3: [It is important to include information about the C-fiber system and its physiology].

Response 3:  [This comment, is in line with the previous one dedicated to the attachment system. Accordingly, as already done for the attachment system, we have included a paragraph entirely dedicated to the physiology of CT fibers (page 14), accompanied by a new table, which summarizes their properties (table 2 at page 14].

Comment 4: [Figure 2 presents a simple diagram that does not illustrate "physiological functional integration" supposed by the legend].

Response 4:  [We fully agree with the observation. For this reason, we have completely changed the structure of the figure (now it has become figure 3 at page 23) according to your suggestion].

Comment 5: [One more problem for readers is the huge amount of abbrevations. It is difficult to remember them all. Some of abbreviations are used only ones, such as International Association for the Study of Pain (IASP),: “Fibromyalgia: Imbalance of Threat and Soothing Systems” (FITSS); central sensitization (CS); prefrontal cortex (PFC); Orbitofrontal Cortex (OFC); insular cortex (IC); Rostral Ventromedial Reticular Structure (RVM) etc. Two times - ventral tegmental area (VTA); Nucleus Accumbens (NAc); periaqueductal gray matter (PAG) etc. Some abbreviations were not explained, such as DHEA-S].

Response 5:  [We recognize that the amount of abbreviations made the manuscript not easily readable. For this reason, we have decided to eliminate many of the acronyms in the list according to the following criteria: the acronym was retained in the text if reported at least eight times or if diffusely used in the clinical and scientific context such us CRPS (e.g. cited in the text only 3 times); we decided to repeat the extended word followed by the acronym between the bracket more than once, for each term potentially unfamiliar to the reader and if the acronym was used in different parts of the manuscript  (e.g. Insecure or Secure Attachment, IA and SA respectively). In this way, we have reduced the original number of the acronyms by half].

Comment 6: [1. To better organize the content, a flowchart can be used. This review would benefit from following the guidelines of the Preferred Reporting Items for Systematic Reviews and Meta-Analyses (PRISMA) https://www.prisma-statement.org/].

Response 6:  [Regarding this point, although we understand and appreciate the attempt to resolve in a way that improve the logical flow of the many concepts reported in this review, we believe that the PRISMA guidelines are not appropriate for this narrative review. However, we realized that by modifying the arrangement of the various subchapter, and adding some others as rightly requested, we believe we have improved the manuscript by making it more fluent. Furthermore, we decided to include a dedicated topic page composed of a list of topics followed by a flow diagram explaining the biological and conceptual flow of the narrative review (Table of contents and figure 1 at pages 3 and 4) also described in the introduction session (please see the added parts written and underlined in red in the revised manuscript)].

Comment 7: [The graphical abstract presents a conceptual overview that would benefit from more detailed explanations in the main text].

Response 7:  [Thank you very much indeed for this comment. To make the manuscript and the flow of the pathophysiological steps described in the graphic abstract clearer, we have added a final paragraph called "conclusions" (page 34) in which we have summarized and explained in a more direct way, our pathophyisiological model so as reported in the graphic abstract (please see the added parts written and underlined in red in the revised manuscript)].

Comment 8: [The text should be reorganized to improve logical flow and readability. Some sentences could be shortened to make the review more concise and clear].

Response 8:  [As already expressed in the previous responses to the other comments, we have added new paragraphs and consequently modified the renaming of the subchapters accordingly as well as the related arrangement. Following your suggestions, as best we could, we rewrote and reformulated some parts of the manuscript (please see the added parts written and underlined in red in the file called “revised manuscript”)].

4. Response to Comments on the Quality of English Language

Point 1:

Response:    (in red)

5. Additional clarifications

[Here, mention any other clarifications you would like to provide to the journal editor/reviewer.]

Round 2

Reviewer 2 Report

Comments and Suggestions for Authors

Thank you for accurately revising the manuscript.
The paper has undergone significant improvements, resulting in a refined and enhanced version of the original work.

All my questions were answered.

Author Response

Thank you for accurately revising the manuscript.
The paper has undergone significant improvements, resulting in a refined and enhanced version of the original work.

All my questions were answered.

Response: Thank you very much indeed for your advices that made the manuscript more readable and accessible also for non specialists in this field of research.

Best Regards

Gianluca Bruti